# Potential Pharmacokinetic Drug–Drug Interaction Between Harmine, a Cholinesterase Inhibitor, and Memantine, a Non-Competitive *N*-Methyl-d-Aspartate Receptor Antagonist

**DOI:** 10.3390/molecules24071430

**Published:** 2019-04-11

**Authors:** Yunpeng Zhang, Shuping Li, Youxu Wang, Gang Deng, Ning Cao, Chao Wu, Wenzheng Ding, Yuwen Wang, Xuemei Cheng, Changhong Wang

**Affiliations:** 1The MOE Key Laboratory for Standardization of Chinese Medicines and The SATCM Key Laboratory for New Resources and Quality Evaluation of Chinese Medicine, Institute of Chinese Materia Medica, Shanghai University of Traditional Chinese Medicine, 1200 Cailun Rood, Shanghai 201203, China; zhangyp1028@163.com (Y.Z.); lishupinghappy@163.com (S.L.); wyxzd1314@163.com (Y.W.); 18930900232@163.com (G.D.); 18616024782@163.com (N.C.); vera105370@163.com (C.W.); 18817385708@163.com (W.D.); wangyuwen92@163.com (Y.W.); chengxuemei1963@163.com (X.C.); 2Shanghai R&D Centre for Standardization of Chinese Medicines, 1200 Cailun Rood, Shanghai 201203, China

**Keywords:** harmine, memantine, drug-drug interaction, pharmacokinetics, Alzheimer’s disease, liquid chromatography-mass spectrometry

## Abstract

Harmine (HAR) is a beta-carboline alkaloid widely distributed in nature. It exhibits psychopharmacological effects of improving learning and memory. However, excessive dose of HAR can cause central tremor toxicity, which may be related to the glutamate system. Memantine (MEM) is a non-competitive *N*-methyl-d-aspartate receptor antagonist. It can be used for the treatment of Alzheimer’s disease and also can block the neurotoxicity caused by glutamate. Therefore, combination of HAR and MEM would be meaningful and the pharmacokinetics investigation of HAR and MEM in combination is necessary. A ultra-performance liquid chromatography tandem mass spectrometry (UPLC-MS/MS) method was established and validated for the simultaneous quantitative determination of MEM, HAR and harmol (HOL), a main metabolite of HAR, in rat plasma after oral administration of HAR and MEM in combination (5.0 mg/kg of MEM combined with 20.0, 40.0, 80.0 mg/kg of HAR). The contents of HAR and HOL were determined after oral administration of HAR (20.0, 40.0 and 80.0 mg/kg), and the content of MEM was determined after oral administration of MEM (5.0 mg/kg). Blood samples were collected from each rat at 0 (pre-dose), 0.08, 0.17, 0.25, 0.33, 0.50, 0.75, 1.0, 2.0, 4.0, 8.0, 12.0 and 24.0 h after administration. The maximum peak concentration (*C_max_*) of MEM was obviously decreased, and the area under the plasma concentration versus time curve from zero to time t (*AUC_(0-t)_*) and mean residence time (*MRT*) were significantly increased after combination with HAR. The *C_max_* and *AUC_(0-t)_* of HAR and its metabolite HOL were increased after combination with MEM. These findings suggested that co-administration of HAR and MEM could extend their residence time in rats, and then might increase the efficacy for treatment of Alzheimer’s disease. Therefore, this study will provide a basis for the rational combined application of HAR and MEM.

## 1. Introduction

Harmine (7-methoxy-1-methyl-9H-pyrido [3,4-β] indole, HAR, structure is shown in Figure 1) is a tricyclic beta-carboline alkaloid originally isolated from seeds of *Peganum harmala* L. HAR is widely distributed in nature, such as various plants, insects, marine creatures, mammalians, human tissues and body fluids [1]. HAR has been traditionally used for medicinal preparations and ritual in the Middle East, Central Asia, South America, Africa, India and China [2,3]. HAR exhibits a wide range of biological activities by inhibiting acetylcholinesterase and monoamine oxidase A (MAO-A) and combining with benzodiazepine, serotonin, imidazoline and opiate receptors [4,5]. HAR possesses a variety of pharmacological effects, particularly in central nervous system. According to previous reports, HAR can enhance short-term memory in old rats [6], and also can effectively ameliorate memory impairments in a scopolamine-induced mouse, which can be used in the treatment of Alzheimer’s disease [7,8].

In view of the toxicity studies of HAR, it is quite similar to 1-methyl-4-phenyl-1,2,3,6- tetrahydropyridine (MPTP) in structure and has been used as model drug to produce one of the main toxin-induced animal models for Parkinson’s disease [9,10,11]. Studies have shown that HAR could lead to central tremor in rats and mice at doses of 200 mg/kg and 38.0 mg/kg, respectively [12,13,14]. Moreover, it could produce rhythmic inferior olive firing, rhythmic purkinje cell complex spike discharges with suppression of simple spikes, and rhythmic discharges of fastigial, interpositus, reticular nucleus, red nucleus, nucleus reticularis tegmentipontis, lateral vestibular nucleus neurons, spinal motoneurons [15,16,17]. It is well known that the stimulation signal is transmitted through glutamate climbing fibers, raising intracellular glutamate levels [18]. Additionally, a large amount of glutamate is released into the extracellular fluid, which has toxic effects on peripheral neurons and then resulting in tremor [19,20]. Therefore, it is speculated that the combination of glutamate receptor antagonists and HAR would attenuate the toxicity of HAR. 

Generally speaking, the over-activation of non-competitive *N*-methyl-d-aspartate (NMDA) receptors is thought to be involved in Alzheimer’s disease [21,22], and which can increase synaptic noise, weaken the detection of related signals such as learning [23]. Memantine (1-amino-3, 5-dimethyladamantane, MEM, structure is shown in Figure 1) is an amantadine derivative, a moderate NMDA receptor antagonist [24,25], which is widely used in treatment of Parkinson’s disease, Alzheimer’s disease and neurogenic bladder dysfunction in spasticity, etc. [22,26]. It is reported that MEM is effective for symptomatic treatment of patients with moderate to severe Alzheimer’s disease [27]. Moreover, the observations in monkeys indicate that glutamate is involved in the pathophysiological cascade of MPTP-induced neuronal cell death, and the glutamate antagonists such as MEM may be able to retard the progression and to improve the symptomatology of Parkinson’s disease [28]. Furthermore, another study shows that MEM can block the neurotoxicity of glutamate without interfering with its physiological actions required for learning and memory [29].

Researches on drug-drug interaction have received much attention. Many drugs have more or less interactions in pharmacokinetics and pharmacodynamics. For example, aspirin combined administration with *Panax notoginseng* saponins can increase absorption of salicylic acid and major active ingredients of *P. notoginseng* saponins [30]. Pretreatment with broad-spectrum antibiotics significantly reduces the plasma exposure of main components of Shaoyao Gancao decoction in rats after oral administration, and this is important for guiding clinical medication [31]. Studies have shown that after oral administration of 40.0 mg/kg HAR, the maximum peak concentration (*C_max_*) of HAR is 67.1 ± 34.3 ng/mL and the elimination half-life (*T_1/2e_*) is 4.73 ± 0.71 h [32]. The pharmacokinetic of MEM in rats has been studied. The *C_max_* of MEM are 89.1 and 370 nM/(mg/kg), and the *T_1/2e_* of MEM are 2.23 and 3.92 h when administered orally with 1.22 and 2.00 mg eq./mL MEM [33]. The pharmacokinetics after the combination of HAR and MEM has not been studied.

Based on previous studies, it is speculated that the combination of MEM and HAR will produce unexpected and expected effects and drug interactions, including pharmacokinetic and pharmacodynamic interactions. Therefore, it is necessary to clarify the pharmacokinetic drug-drug interactions of HAR and MEM first. The aim of this study is to develop a sensitive and simple ultra-performance liquid chromatography tandem mass spectrometry (UPLC-MS/MS) method for measuring MEM, HAR and harmol (HOL, structure is shown in Figure 1), one of the main metabolites of HAR [34], with tacrine (9-amino-1,2,3,4-tetrahydroacridine hydrochloride hydrate, structure is shown in Figure 1) as an internal standard (IS), and the validated method was applied to explore the pharmacokinetics drug-drug interactions of HAR and MEM in combination. The present results will provide significant information for HAR and MEM administration alone or in combination, and will be helpful for their clinical practice.

## 2. Results

### 2.1. Optimization of UPLC-MS/MS Conditions

The optimization of UPLC-MS/MS conditions was done by infusing the standard solutions of MEM, HAR, HOL and IS directly into the electrospray ionization (ESI) source. The signal intensities of analytes and IS were higher and more stable when positive ESI mode was selected. The precursor and product ions of analytes and IS were selected in positive mode. The most sensitive multiple reaction monitoring (MRM) transitions used for HAR, MEM, HOL and IS were 213.2/170.0, 180.2/163.1, 199.2/131.1 and 199.2/144.1, respectively. The product ion mass spectra of HAR, MEM, HOL and IS are shown in Figure 2. MS specific optimized critical parameters of analytes and IS which ascertain reproducible responses were tabulated in Appendix A. Methanol was used as the organic phase because it provided higher responses and lower background noise than acetonitrile. The sensitivity and peak symmetry of analytes and IS were remarkably improved by adding 0.1% formic acid to the mobile phase. 

### 2.2. Optimization of the Pretreatment Procedure

The pretreatment procedure was an important step for a reliable and accurate UPLC-MS/MS assay. The protein was more completely precipitated when the ratio of plasma to methanol was 1:4 (*v*/*v*). A solvent effect of the analytes and IS could be effectively reduced adding an appropriate amount of 0.1% formic acid before injection. 

### 2.3. Method Validation

#### 2.3.1. Selectivity and Carry-Over

The typical MRM chromatograms of blank plasma sample, blank plasma sample spiked with MEM, HAR and HOL at lower limit of quantification (LLOQ) and IS (40.0 ng/mL), and the IS-spiked plasma sample obtained at 15 min after the co-administration of medium dose (40.0 mg/kg) HAR with 5.0 mg/kg MEM were shown in Appendix A. No obvious interference from endogenous matrix was observed for HAR, MEM, HOL and IS by MRM mode. The retention time of MEM, HAR, HOL and IS was 3.58, 2.78, 1.95, and 2.30 min, respectively. No peak was observed at the retention time of analytes or IS in the chromatogram of the blank sample analyzed after the injection of upper limit of quantification (ULOQ) sample, indicating the absence of carry-over.

#### 2.3.2. Linearity, the Low Limit of Detection (LLOD) and LLOQ

The calibration curves shown good linearity within the concentration range of 0.40–400 ng/mL for MEM, HAR and HOL, respectively, in rat plasma. Typical equations of the calibration curves for MEM, HAR and HOL were: *y* = 0.550 *x* + 6.38 × 10^−3^ (*r^2^* = 0.997), *y* = 3.23 *x* + 2.94 × 10^−2^ (*r^2^* = 0.998), and *y* = 4.05 *x* − 1.85 × 10^−3^ (*r^2^* = 0.998), respectively. The LLOD values for analytes were 0.10 ng/mL, and their LLOQ values were 0.40 ng/mL with acceptable limits of accuracy and precision. The above detailed data were shown in Appendix A. 

#### 2.3.3. Within-Run and Between-Run Precision

Within-run and between-run precision of the developed method was validated by determining the quality control (QC) samples at five concentrations (0.40, 1.00, 100, 300, and 400 ng/mL) in plasma. Appendix A summarized the results of within-run and between-run precision for MEM, HAR and HOL. The intra-run coefficient of variation (CV)% as within-run imprecision was between 1.84% and 8.86% at the five concentrations. Inaccuracy expressed as intra-run relative error (RE)% was below 17.6% at the LLOQ, and 9.90% at other concentrations. The inter-run CV% as between-run imprecision did not exceed 6.02% and the inter-run RE% as inaccuracy expressed was no more than 10.1%. These results were in compliance with Food and Drug Administration (FDA) requirements, and the biological sample analysis method was proved to be consistent and precise at different sample concentrations. 

#### 2.3.4. Matrix Effect and Extraction Recovery

The matrix effect and extraction recovery values of MEM, HAR, HOL and IS were summarized in Appendix A. The matrix effect values for analytes ranged within 96.4–102% and the mean ± standard deviation (SD) values were below 7.50% at three QC levels (n = 6). In addition, the matrix effect values of IS were 99.0 ± 1.68% (n = 6). Therefore, the ion enhancement or suppression of plasma components to analytes and IS were negligible in this method. The extraction recoveries of the three analytes were between 97.2% and 105% at the three evaluated concentrations and the SD values were below 6.34% (n = 6). The extraction recoveries of IS were 103 ± 0.663% (n = 6). Therefore, the recoveries of analytes and IS satisfied the requirements of the bioanalytical method. 

#### 2.3.5. Stability

The results of the stability experiments under various conditions were summarized in Appendix A. The analytes in plasma samples were stored at ambient temperature for 4 h. The SD values were within 0.01–9.79% and the CV% values were no more than 3.75% at low, medium and high QC levels (n = 6). The three QC levels were stored at 4 °C for 24 h, for one month under −20 °C, and repeated three freeze-thaw cycles. The SD values were between 0.03% and 16.50% and the CV% did not exceed 6.68%. The analytes and IS stock solutions were verified at room temperature for 24 h, at 4 °C for three days and −20 °C for one month. In summary, the stability of analytes and IS in plasma sample and standard stock solutions met the requirements of a routine pharmacokinetic study. 

### 2.4. Pharmacokinetics Studies

The validated UPLC-MS/MS method was successfully applied to the in vivo pharmacokinetics studies of MEM, HAR and HOL in the rat plasma after oral administration of MEM and HAR alone and in combination. The mean plasma concentration time curves were shown in Figure 3, Figure 4 and Figure 5. The pharmacokinetic parameters of MEM, HAR and HOL based on non-compartmental method were presented in Table 1, Table 2 and Table 3. 

After oral administration of MEM at a dose of 5.0 mg/kg (MEM group), the plasma concentration versus time curve of MEM rose rapidly, the maximum peak concentration (*C_max_*) of MEM was 84.8 ± 25.8 ng/mL when time of maximum plasma concentration (*T_max_*) reached to 1.31 ± 0.579 h, and then it rapidly declined with elimination rate constant (*K_e_*) of 0.261 ± 0.0282 /h and elimination half-life (*T_1/2e_*) of 2.69 ± 0.306 h, and no MEM was detected after 24 h of administration. However, after co-administration of MEM (5.0 mg/kg) and HAR (20.0, 40.0 and 80.0 mg/kg, HAR-L + MEM, HAR-M + MEM and HAR-H + MEM group), the mean plasma concentration time curves of MEM rose slowly (in HAR-L + MEM and HAR-H + MEM groups) and the *C_max_* of MEM decreased to 50.5 ± 7.59, 74.9 ± 15.0 and 74.3 ± 26.6 ng/mL with *T_max_* of 3.13 ± 2.23 and 4.00 ± 1.85 h. The *C_max_* of MEM was 74.9 ± 15.0 ng/mL with *T_max_* of 1.56 ± 1.11 h in HAR-M + MEM group. The *C_max_* values of three co-administration groups were lower than that of a single oral dose of MEM. Additionally, a small amount of MEM could still be detected within 24.0 h after co-administrations. For *K_e_*, the group of HAR-L + MEM was significantly lower than that of MEM group (*p* < 0.01), and for *T_1/2e_*, the HAR-L + MEM group was higher than the MEM group (*p* < 0.05). Statistical analysis shown that there were significant differences in *MRT* of HAR-L + MEM and HAR-M + MEM groups compared with MEM group (*p* < 0.001). The *T_max_*, area under the plasma concentration versus time curve from zero to time t (*AUC_(0-t)_*) and mean residence time (*MRT)* values of HAR-H + MEM group were much higher than those of MEM group (*p* < 0.01). The *AUC_(0-t)_* was dose-dependently elevated in the three co-administered groups, and the *AUC_(0-t)_* in HAR-H + MEM group was significantly higher than that in HAR-M + MEM group (*p* < 0.05). Compared to the single group, the ratio of clearance to bioavailability (*CL/F*) of co-administered groups decreased dose-dependently with the increasing dosage of HAR (*p* < 0.01, *p* < 0.01, *p* < 0.001) (Figure 3, Table 1).

The mean plasma concentration time curves and the pharmacokinetic parameters of low, medium and high doses of HAR groups (HAR-L, HAR-M and HAR-H) and three doses of HAR combined with MEM groups were presented in Figure 4 and Table 2. The *C_max_* values of HAR in co-administration groups were higher than those in HAR group (HAR-L + MEM: *p* < 0.001; HAR-H + MEM: *p* < 0.05), and there is a dose-dependent increase in the middle and high-dose administration groups and in the middle and high-dose co-administered groups (*p* < 0.001). The *K_e_* of the HAR-M + MEM group was lower than that of the HAR-M group (*p* < 0.05), while the *T_1/2e_* shown the opposite trend in both groups (*p* < 0.05). The *AUC_(0-t)_* values of co-administration groups were higher than those of HAR group (HAR-L + MEM and HAR-H + MEM: *p* < 0.05). The *AUC_(0-t)_* was dose-dependently elevated in the three HAR groups, and the *AUC_(0-t)_* in HAR-H group was significantly higher than that in HAR-M group (*p* < 0.001). The *CL/F* of single groups was dose-dependently increase, and the *CL/F* in HAR-H group was significantly higher than that in HAR-M group (*p* < 0.01). Compared to the single groups, the *CL/F* of co-administered groups decreased, respectively (HAR-M + MEM: *p* < 0.05). No significant difference was observed in distribution rate constant (*K_d_*), absorption rate constant (*K_a_*), distribution half-life (*T_1/2d_*), absorption half-life (*T_1/2a_*), *T_max_*, apparent volume of distribution (*V_d_*) and *MRT* between co-administration and single groups. There was no significant difference in *C_max,_ AUC_(0-t)_* and *CL/F* between the middle-dose groups and the low dose groups.

In the low dose groups of single and co-administration, the concentration of HOL was lower than LLOQ in some plasma samples, and the complete plasma concentration time curves could not be drawn. Therefore, Figure 5 and Table 3 only shown the plasma concentration time curves and pharmacokinetic parameters of medium and high doses of HAR and co-administration groups. The plasma concentration of HOL in HAR-H + MEM group was a little different. This may be the cause of hepatic and intestinal recirculation. In the drug-administered group, the *C_max_* and *AUC_(0-t)_* in the high-dose groups were greater than those in the middle-dose group. The *C_max_* of HOL in HAR-M + MEM group was higher than that in HAR-M group, and the *T_1/2e_* and *MRT* of HOL in the HAR-H + MEM group were larger than those in the HAR-H group (*p* < 0.05). The *Ke* of HOL in the HAR-H + MEM group was much lower than that in the HAR-H group (*p* < 0.01), while the *AUC_(0-t)_* of HOL shown the opposite trend in both groups (*p* < 0.01). The *CL/F* of HAR-H + MEM group was significantly lower than that of HAR-M + MEM group (*p* < 0.001). The *CL/F* of HOL in the HAR-H + MEM group was significantly lower than that in the HAR-H group (*p* < 0.001). There were no significant changes were observed in *K_a_*, *T_1/2a_*, *C_max_* and *T_max_* of HOL between single and co-administration groups. 

## 3. Discussion

With the rapid growth of the global aging population, Alzheimer’s disease has become an important public health challenge [35]. In recent years, people pay more attention to Alzheimer’s disease, and more and more studies have been conducted on it. The “choline hypothesis” has been proposed, which suggests that the loss of cholinergic function in the central nervous system is significantly associated with the cognitive abilities decline of Alzheimer’s disease [36]. In addition to low cholinergic capacity, glutamate metabolism also plays an important role in brain cognitive function [37]. Studies have shown that glutamate over-activation of postsynaptic NMDA receptors may lead to neuron damage [37]. Most treatments for Alzheimer’s disease are based on disorders of the neurotransmitters acetylcholine and glutamate. Acetylcholinesterase inhibitors, such as donepezil, galantamine, and rivastemine have been approved for the treatment of mild to moderate Alzheimer’s disease. Tacrine, the first acetylcholinesterase inhibitor on the market, has serious side effects such as liver toxicity and has been replaced by other available acetylcholinesterase inhibitor [38]. 

On the one hand, HAR can ameliorate impaired memory by enhancement of cholinergic neurotransmission via inhibiting the activity of acetylcholinesterase [7,8]. On the other hand, HAR can increase intracellular glutamate levels through glutamate climbing fibers, causing central toxic tremor [16,19,39]. The literature reports that MEM can reduce glutamate content and attenuate central toxicity through the glutamate system [28]. To explore whether the combination administration of HAR and MEM has a synergistic effect, a pharmacokinetic drug–drug interaction experiment was conducted firstly for co-administration of HAR and MEM. 

In this study, The *C_max_* and *T_1/2e_* of MEM were consistent with previous studies [33]. Compared with the MEM group, after combined administration of HAR and MEM, the *AUC_(0-t)_* and *MRT* of MEM was significantly increased, and *CL/F* of MEM was significantly decreased after combination administration with different doses of HAR in a dose-dependent manner. This indicated that HAR could slow down the metabolism of MEM, increase the residence time of MEM, and expand the safety window of MEM, and which is helpful for treating Alzheimer’s disease and Parkinson’s disease, etc. That is to say, the elimination of MEM was weakened. This result is consistent with previous studies that HAR, HOL and MEM are all cleared by the kidneys and will suppress each other [34,40]. The *K_a_* of MEM was decreased after combined administration, meaning that HAR slowed the absorption of MEM. It is shown that HAR is a substrate for the absorption transporter organic cation transporters (OCTs/OCTNs) and organic anion transporting polypeptides (OATPs) in human colon carcinoma and Madin-Darby canine kidney cells [32], and MEM is a substrate for the absorption transporter OCT2 and OATP2B1 in Madin-Darby canine kidney and human embryonic kidney cells [33,41]. It is speculated that HAR might compete with MEM for the same absorption transporter OCTs and OATPs in rat, which will reduce the absorption of MEM after co-administration. Additionally, the results shown that the *C_max_* of MEM after co-administration was lower than that of MEM alone, it also proved the above speculation. When medium and high doses of HAR combined with MEM, the *C_max_* of MEM was higher than that of low dose HAR combined with MEM. This may be due to the administration of a large dose of HAR could induce compensatory effects and induce the activity of uptake transporters in rat, and then result in an increase absorption. According to these findings, the combination administration of MEM and HAR maybe more helpful for treating Alzheimer’s disease and decreasing side effects caused by excessive MEM. 

The *C_max_* of HAR was consistent with previous studies, and the *T_1/2e_* of HAR was lower than previous studies [32]. The *C_max_* and *AUC_(0-t)_* of HAR and HOL in the combined groups were higher than those in the HAR group, while the *CL/F* of HAR and HOL shown the opposite trend. Studies have shown that the HAR is mainly metabolized by cytochrome P450 proteins (CYP)1A2, CYP2D6 and CYP3A4 in human and rat liver microsomes [42,43,44], and MEM is a strong inhibitor of CYP2D6, and a weak inhibitor of CYP1A2 and CYP3A4 in rat liver microsomes [45]. HAR, HOL and MEM could be eliminated by the kidneys [34,40]. The present results suggested that MEM might inhibit the metabolic enzymes and compete for the eliminate organs with HAR and HOL. Thence, the exposure of HAR and HOL in rats was increased after co-administration, and it might be benefit for increasing the efficacy of HAR in improving learning and memory. 

After the combination of HAR and MEM, the elimination of both drugs was slowed down, it would result in enhancing the efficacy of the treatment for Alzheimer’s disease. In addition, HAR has a variety of other pharmacological effects, such as lowering systemic arterial blood pressure and raising pulse pressure, increasing brain-derived neurotrophic factor protein levels in rat hippocampus to prevent depression, and treating cancer [46,47,48]. These results of this experiment suggest that MEM may improve the above effects of HAR or reduce the dose of HAR to achieve the same efficacy. In the clinical use of MEM, we can also reduce the amount of MEM and improve the efficacy by combination therapy. However, the current findings do not indicate that MEM can weaken the toxicity of HAR. This may be due to the high dose of HAR (80.0 mg/kg) is lower than its toxic dose, and which needs further experimental research to explore. On the other hand, drug–drug interaction is a complex process and the combination of HAR and MEM may have a certain effect on changes in glutamate, dopamine, choline, inflammatory factors and reactive oxygen species. Hereafter, more experiments should be designed to explore the effects of combination drugs on the glutamate system to clarify whether MEM can attenuate the central toxicity of HAR.

The pharmacokinetic parameters may vary from species to species and the interspecies differences may origin from drug absorption, distribution, metabolism and elimination. Biotransformation is a major factor accounting for species differences in the disposition of drug [49]. Animal models, rats in particular, are commonly used to predict the metabolism and toxicity of new human drug candidates. However, it is important to realize that humans differ from animals in metabolic enzyme isoform composition, expression and catalytic activity and even small changes in the amino acid sequences of these enzymes can give rise to profound differences in substrate specificity and catalytic activity [50]. These magnify a complexity in the extrapolation of animal’s data to humans. 

Comparative metabolism is relevant when identifying animal models for humans that have an appropriate similarity, but none of the investigated species matches all the typical CYP450 activities as described in humans [49]. With the aim of establishing the best animal model for human CYP450-related research, Li et al. undertook a comprehensive comparative investigation of the enzyme activities and kinetic parameters of HAR for individual CYP450 activities in liver microsomes or S9 fractions derived from rat, mouse, pig, bull, sheep, camel, rabbit, dog, monkey, guinea pig and human. The overall conclusion of this comparison is that none of the investigated species matches all the typical CYP450 activities as described in humans. Additionally, dogs and humans show considerable similarity in the metabolic profiles and catalytic processes of HAR [51]. It suggests that the pharmacokinetic drug–drug interaction between HAR and MEM would need to be carried out on other different animal models, such as dogs, monkeys and even clinical trials in the future study. This can be more comprehensive, can better guide clinical medication. 

By the way, it should be given considerable attention that the nomenclature and function of CYP metabolic enzymes are different among different species. For instance, the amino acid sequences of CYP2A1 and CYP2A2 in rats and CYP2A6 in human are highly homologous [52], CYP2D22 of mice is homologous to CYP2D6 of humans [53], and CYP2D21 of pig is highly homologous to CYP2D6 of human [54]. 

In addition, drug transporters also play an important role in the absorption, distribution, and excretion of many drugs. There are significant differences between rodents, dog, monkey and human in the substrate specificity, tissue distribution, and relative abundance of transporters [55]. These differences complicate cross-species extrapolations, which is crucial when attempting to predict human pharmacokinetics of drug candidates and assess risk for drug–drug interactions. However, quantitative knowledge of species differences of transporters, especially at the protein and functional level, is still limited. Further increased understanding of species differences in transporter expression and functional activity is needed in order to translate findings from preclinical species to humans, which will improve the capability to predict drug–drug interactions characteristics of drug candidates in humans.

## 4. Materials and Methods

### 4.1. Reagents and Materials

HAR with purity of >98.0% was isolated by high performance liquid chromatography (HPLC) from seeds of *P. harmala* and then identified by mass spectrometric (MS) and nuclear magnetic resonance (NMR) in our laboratory [56]. MEM hydrochloride (purity ≥ 98.0%) was isolated from Source Leaf Creature (Shanghai, China). HOL (purity ≥ 98.0%) was purchased from Wako Pure Chemical Industry Co., Ltd. (Osaka, Japan). Tacrine and heparin sodium were obtained from Sigma-Aldrich Co. (St. Louis, MO, USA). HPLC grade methanol and formic acid were purchased from Fisher Scientific Co. (Santa Clara. CA, USA). Ultrapure water was obtained using a Milli-Q Academic System (Millipore, Corp., Billerica, MA, USA). All other reagents and solvents were of either analytical or HPLC grade. Drug-free rat plasma containing heparin as anticoagulant was collected from adult healthy male Sprague-Dawley rats. 

### 4.2. Animals and Ethics Statement

Fifty-six pathogen-free male Sprague-Dawley adult rats (weighing within the range of 200–250 g) were provided by the Experimental Animal Center of Shanghai University of Traditional Chinese Medicine (Permit Number: SCXK (Hu) 2013-0016). The rats were raised under an environmentally controlled room for seven days before starting the experiments. The rats were housed with free access to food and water and maintained on a 12 h light and dark cycle at 60% to 65% relative humidity and environmental temperature (25 ± 1 °C). All operations related to animals were in accordance with the regulations for animal experimentation issued by the State Committee of Science and Technology of the People’s Republic of China on 14 November 1988 and approved by the Animal Ethics Committee of Shanghai University of Traditional Chinese Medicine (No. PZSHUTCM18122111, Approval date: 21 December 2018). 

### 4.3. Instruments and Conditions

#### 4.3.1. Liquid Chromatography

The liquid chromatography (LC) system consisted of an Agilent 1290 HPLC system with auto-sampler (Agilent Technologies, Santa Clara, CA, USA). The analytes were separated on an ACQUITY UPLC BEH C_18_ column (2.1 mm × 50 mm, 1.7 μm, Waters, Milford, USA). The column was eluted with a gradient mobile phase of methanol (A) and 0.1% formic acid in deionized water (B): 0–1.0 min, 9.0% to 30.0% A; 1.0–2.0 min, 30.0% to 40.0% A; 2.0–4.0 min, 40.0% to 90.0% A; 4.0–5.0 min 90.0% A; 5.0–5.01 min, 90.0% to 9.0% A; 5.01–7.0 min, 9.0% A for equilibration of the column. The flow rate was 0.3 mL/min. The column and sample-tray temperature were kept at 40 °C and 4 °C, respectively. The injection volume was 5.0 μL using a partial loop with needle over fill mode. 

#### 4.3.2. Mass Spectrometric Conditions 

MS detection was performed using an Agilent 6410 triple stage quadrupole mass spectrometer equipped (Agilent Technologies, Santa Clara, CA, USA) with an ESI ion source in positive ionization mode. Tune parameters were optimized and established as follows: gas temperature, 350 °C; gas flow, 10.0 L/min; nebulizer pressure, 40.0 psi; capillary voltage, 4 × 10^3^ V. MS acquisition of MEM, HAR, HOL and IS was performed in electrospray positive ionization MRM mode. The compound dependent parameters used for analysis were summarized in Appendix A. 

### 4.4. Preparation of Standard Solutions, Calibration Standards and Quality Control Samples 

The standard stock solutions of MEM, HAR and HOL with the concentration of 100.3, 101.5, 100.4 μg/mL were prepared by dissolving proper amount of each standard in 25.0 mL of methanol, respectively. Mixed working stocks of MEM, HAR and HOL were prepared by serial dilution in methanol to give a series of standard solutions with different concentrations. An IS working solution (100.3 μg/mL) was also prepared by dissolving proper amount of tacrine in 25.0 mL of methanol. The stock solutions of MEM, HAR and HOL were serially diluted with the initial mobile phase to obtain three working solutions containing MEM, HAR and HOL with concentrations ranging within 0.40–400 ng/mL and IS with a concentration of 40.0 ng/mL for the standard curves. QC samples were prepared independently in the same steps at low (1.0 ng/mL), medium (100 ng/mL) and high (300 ng/mL) levels for MEM, HAR and HOL by using different standard working solutions. All the solutions were stored at 4 °C. 

### 4.5. Sample Preparation

A rapid and convenient precipitation method was used to prepare the plasma samples. A 100 μL aliquot of plasma sample was spiked with 100 μL of IS solution and 300 μL of methanol in a 1.5 mL centrifuge tube; subsequently, the mixtures were vortex-mixed for 30.0 s and then centrifuged at 1.5 × 10^4^× *g* for 10 min at 4 °C. A 100 μL aliquot of supernatant was combined with 25.0 μL of 0.1% formic acid water in a clean centrifuge tube and then vortex-mixed for 30 s. A 5.0 μL aliquot of mixture was injected into the UPLC-MS/MS system for analysis. 

### 4.6. Method Validation

The bioanalytical method was fully validated according to the U.S. Food and Drug Administration Bioanalytical Method Validation Guidance, European Medicines Agency Guideline on Bioanalytical Method Validation and other related guidelines with respect to selectivity and carry-over, linearity, LLOD and LLOQ, within-run and between-run precision, matrix effect, extraction recovery and stability [42]. 

#### 4.6.1. Selectivity and Carry-Over

The selectivity was evaluated by comparing the MRM chromatograms of blank plasma, blank plasma spiked with MEM, HAR and HOL at LLOQ and IS, and IS-spiked plasma sample after a co-administration of HAR at a dose of 40.0 mg/kg and MEM at a dose of 5.0 mg/kg. Carry-over test was performed in triplicate by injecting a blank crude preparation sample extract followed by immediate injection of an extract of sample from the ULOQ (400 ng/μL) along with IS. The carry-cover in the blank sample after injection of ULOQ should not exceed 20.0% of the LLOQ, and should not exceed 5.0% of the IS. 

#### 4.6.2. Linearity, LLOD, and LLOQ

Calibration curves were prepared within the range of 0.40–400 ng/mL in five replicates on each validation run, and the standard curves were drawn by the peak area ratio (analytes/IS, *y*) versus the concentration ratio (analytes/IS, x). The LLOD was defined as the lowest concentration that produced a peak distinguished from the background noise (a minimum *S/N* ratio of 3:1) for the analytes. The LLOQ was expressed as the lowest concentration giving a signal-to-noise (*S/N*) ratio of at least 10-fold and on the calibration curve with an acceptable accuracy and precision (relative standard deviation, RSD: below 20.0%). 

#### 4.6.3. Within-Run and Between-Run Precision 

The precision of the analytical method was defined as the CV, which including within-run precision and between-run precision. Precision was determined by performing replicate analysis of QC samples (n = 6) at five concentrations. The within-run precision was assessed by repeating the analysis of the standard six times during a single analytical run, and the between-run precision was evaluated by repeating the analysis of the standard six times during three consecutive days with six analytical runs. The CV and RE of within-run and between-run precision should not exceed 15.0% for the quality control samples, except for the CV and RE of LLOQ which should not exceed 20.0%.

#### 4.6.4. Matrix Effect and Extraction Recovery

The matrix effect of MEM, HAR, HOL and IS was investigated at low, medium and high QC levels, using six batches of blank matrix from individual rat. Calculating the peak area ratios of the blank matrix after extraction spiked with MEM, HAR, HOL and IS to the peak area of pure solution containing the analytes at the same levels. The CV of the IS-normalized matrix factor calculated from the six batches of matrix should not higher than 15.0%. The extraction recoveries were inspected at three QC concentrations with six replicates. The recovery was determined as the ratio of measured concentration to actual concentration. The concentrations were calculated by calibration curves. 

#### 4.6.5. Stability

The stability of MEM, HAR and HOL in plasma sample was assessed by analyzing replicates (n = 6) at low, medium and high concentrations. For long-term stability, the QC samples were stored at −20 °C for one month. Samples were stored for 24 h in the auto-sampler at 4 °C, for 4 h at ambient temperature (25 ± 1 °C). The freeze-thaw stability was detected after thawing at ambient temperature and freezing at −20 °C for three cycles. At each QC sample level, the imprecision should not to exceed 15.0%. 

### 4.7. Pharmacokinetic Study

Fifty-six pathogen-free male adult rats were randomly divided into seven groups with eight rats in each group for pharmacokinetic study. Rats fasted for 12 h prior to the study and had free access to water. One group was intragastrically administered with MEM hydrochloride 5.0 mg/kg dissolved in ultrapure water, namely, MEM group. Three groups were oral administered with HAR at doses of 20.0, 40.0 and 80.0 mg/kg, namely, low: HAR-L, medium: HAR-M, and high: HAR-H, respectively. Additionally, the other three groups were administered with HAR at doses of 20.0, 40.0 and 80.0 mg/kg and MEM at dose of 5.0 mg/kg, namely, HAR-L + MEM, HAR-M + MEM and HAR-H + MEM, respectively. Blood samples of approximately 0.25 mL were collected from each rat and transferred into heparinized tubes at 0 (pre-dose), 0.08, 0.17, 0.25, 0.33, 0.5, 0.75, 1.0, 2.0, 4.0, 8.0, 12.0 and 24.0 h after administration. The blood samples were immediately centrifuged at 4.0 × 10^3^× *g* at 4 for 10 min, and 100 μL of the supernatant plasma was transferred into another new 1.5 mL centrifuge tube and stored at −20 °C until UPLC-MS/MS analysis. 

### 4.8. Statistical Analysis

All standard curves and quantitative data were processed by Agilent Mass Hunter Quantitative Analysis Workstation Software. Pharmacokinetic parameters and the experimental data were expressed as the mean ± SEM. The blood sample concentration versus time curves of MEM, HAR and HOL were drawn by GraphPad Prism 5. All the pharmacokinetic parameters were processed using the non-compartmental pharmacokinetics data analysis software program PK solutions 2.0^TM^ (Summit Research Services, USA). The calculated pharmacokinetic parameters of MEM, HAR and HOL were: *K_a_*, *K_d_*, *K_e_*, *T_1/2a_*, *T_1/2d_*, *T_1/2e_*, *MRT*, apparent volume of distribution (*V_d_*) and *CL/F*. The *C_max_* and *T_max_* were acquired from the observed concentration versus time data. The *AUC_(0-t)_* was obtained by the linear/logarithmic trapezoidal rule, and the area under the plasma concentration versus time curve from zero to infinity (*AUC_(0-∞)_*) was calculated by means of the trapezoidal rule with extrapolation to infinity with a terminal elimination rate constant. The Pearson correlation coefficient analysis was processed by IBM SPSS Statistics Version 21. A statistical analysis was performed using analysis of variance with α = 0.05 as the minimal level of significance.

## 5. Conclusions

A sensitive, rapid and accurate UPLC-MS/MS method for simultaneous quantitative determination of MEM, HAR and HOL was established and successfully applied for the drug–drug pharmacokinetics interaction study of HAR and MEM. After co-administration of HAR and MEM, the elimination of the HAR, HOL and MEM was slowed down, and their residence time was increased in rats. Combination of HAR and MEM may help to extend the therapeutic effects of HAR and MEM on Alzheimer’s disease and others. Overall, the present study is most helpful for interpreting the drug–drug interaction of HAR and MEM, and will provide valuable information for the follow-up study and clinical medication in combination of HAR and MEM. 

## Figures and Tables

**Figure 1 molecules-24-01430-f001:**
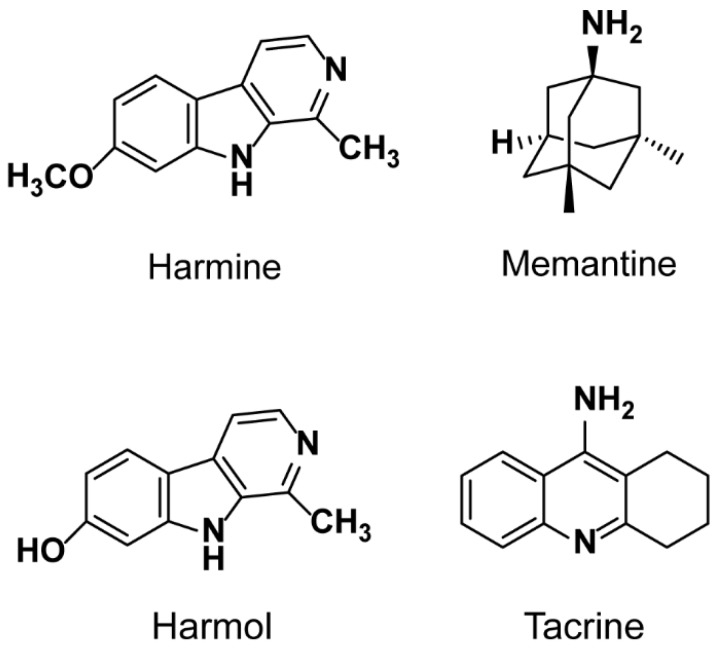
Chemical structures of harmine (HAR), memantine (MEM), harmol (HOL) and tacrine (IS).

**Figure 2 molecules-24-01430-f002:**
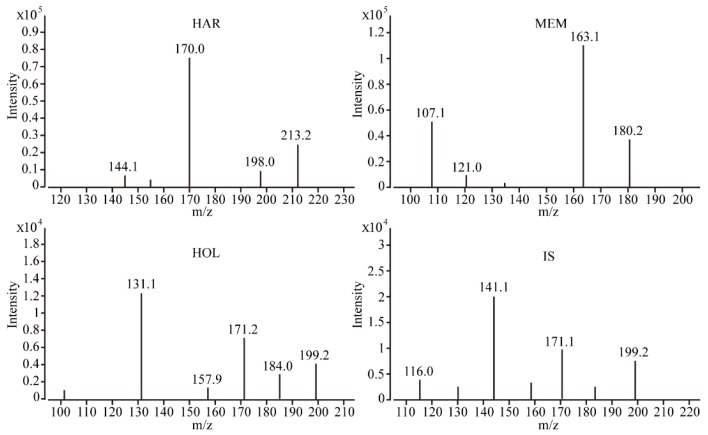
Product ion mass spectra of harmine (HAR), memantine (MEM), harmol (HOL) and tacrine (IS).

**Figure 3 molecules-24-01430-f003:**
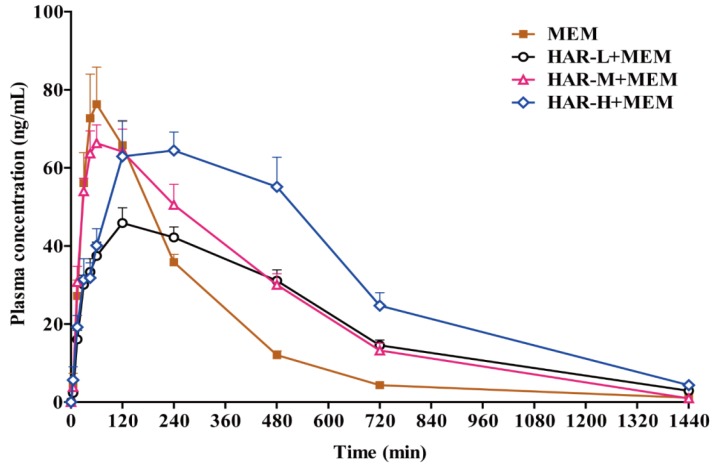
Mean plasma concentration time curves of MEM in rats plasma after single administration of MEM at a dose of 5.0 mg/kg and oral administration of low (20.0 mg/kg), medium (40.0 mg/kg) and high (80.0 mg/kg) doses of HAR combined with MEM (5.0 mg/kg) (n = 8, mean ± SEM).

**Figure 4 molecules-24-01430-f004:**
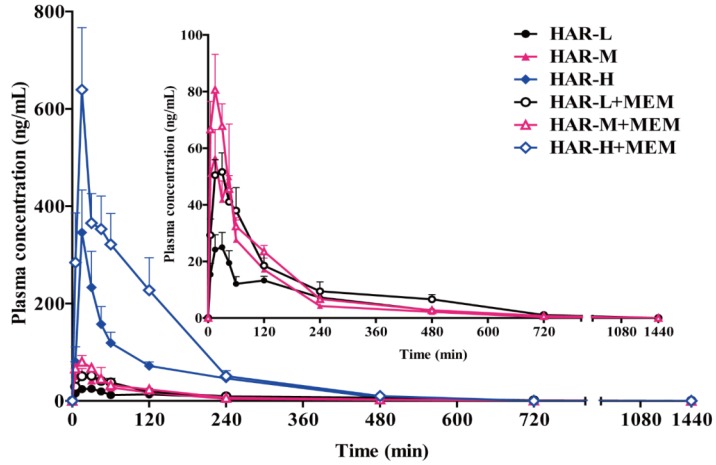
Mean plasma concentration time curves of HAR in rats plasma after oral administration of low (20.0 mg/kg), medium (40.0 mg/kg) and high (80.0 mg/kg) doses of HAR and low (20.0 mg/kg), medium (40.0 mg/kg) and high (80.0 mg/kg) doses of HAR combined with MEM (5.0 mg/kg) (n = 8, mean ± SEM).

**Figure 5 molecules-24-01430-f005:**
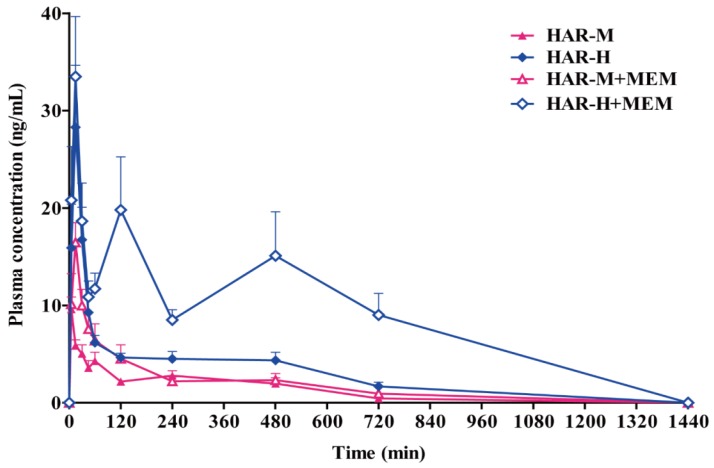
Mean plasma concentration time curves of HOL in rats plasma after oral administration of medium (40.0 mg/kg) and high (80.0 mg/kg) doses of HAR and medium (40.0 mg/kg) and high (80.0 mg/kg) doses of HAR combined with MEM (5.0 mg/kg) (n = 8, mean ± SEM).

**Table 1 molecules-24-01430-t001:** Pharmacokinetics parameters of MEM in rats after single administration of MEM at a dose of 5.0 mg/kg and oral administration of low (20.0 mg/kg), medium (40.0 mg/kg) and high (80.0 mg/kg) doses of HAR combined with MEM (5.0 mg/kg) (n = 8, mean ± SD).

Pharmacokinetics Parameters	MEM	HAR-L + MEM	HAR-M + MEM	HAR-H + MEM
*K_e_* (/h)	0.261 ± 0.0282	0.191 ± 0.0508 **	0.215 ± 0.0728	0.203 ± 0.0723
*K_d_* (/h)	0.548 ± 0.669	0.320 ± 0.0934	0.488 ± 0.262	0.485 ± 0.202
*K_a_* (/h)	1.73 ± 0.848	0.595 ± 0.315 **	1.08 ± 1.05	0.481 ± 0.203 **
*T_1/2e_* (h)	2.69 ± 0.306	3.96 ± 1.44 *	3.62 ± 1.42	4.27 ± 3.03
*T_1/2d_* (h)	3.08 ± 2.37	2.42 ± 1.06	1.67 ± 0.560	1.61 ± 0.529
*T_1/2a_* (h)	0.482 ± 0.202	1.37 ± 0.468 ***	0.945 ± 0.398 **	1.61 ± 0.459 *** ^bb^
*C_max_* (ng/mL)	84.8 ± 25.8	50.5 ± 7.59 **	74.9 ± 15.0^b^	74.3 ± 26.6
*Tmax* (h)	1.31 ± 0.579	3.13 ± 2.23	1.56 ± 1.11	4.00 ± 1.85 ** ^b^
*AUC_(0-t)_* (ng*h/mL)	349 ± 46.2	392 ± 55.2	472 ± 105 *	603 ± 149 ** ^b^
*AUC_(0-_**_∞)_*(ng*h/mL)	367 ± 4.87	477 ± 75.1 **	547 ± 142 **	781 ± 285 ** ^b^
*MRT* (h)	4.04 ± 0.808	7.45 ± 1.51 ***	6.25 ± 1.25 ***	8.50 ± 3.74 **
*V_d_* (L/kg)	53.8 ± 10.6	60.1 ± 19.3	48.4 ± 16.4	38.3 ± 13.1 *
*CL/F* (L/h/kg)	13.8 ± 1.50	10.7 ± 1.79 **	9.70 ± 2.57 **	7.10 ± 2.29 *** ^b^

* *p* < 0.05; ** *p* < 0.01; *** *p* < 0.001 (Combined administration compared with MEM). ^b^
*p* < 0.05; ^bb^
*p* < 0.01 (HAR-M + MEM compared with HAR-L + MEM and HAR-H + MEM compared with HAR-M + MEM).

**Table 2 molecules-24-01430-t002:** Pharmacokinetics parameters of HAR in rats plasma after oral administration of low (20.0 mg/kg), medium (40.0 mg/kg) and high (80.0 mg/kg) doses of HAR and low (20.0 mg/kg), medium (40.0 mg/kg) and high (80.0 mg/kg) doses of HAR combined with MEM (5.0 mg/kg) (n = 8, mean ± SD).

Pharmacokinetics Parameters	HAR-L	HAR-M	HAR-H	HAR-L + MEM	HAR-M + MEM	HAR-H + MEM
*K_e_* (/h)	0.428 ± 0.226	0.477 ± 0.277	0.419 ± 0.271	0.312 ± 0.115	0.174 ± 0.0543 *	0.456 ± 0.407 ^b^
*K_d_* (/h)	1.62 ± 2.43	2.06 ± 2.79	1.11 ± 1.11	1.40 ± 1.28	0.743 ± 0.741	0.921 ± 0.559
*K_a_* (/h)	1.66 ± 2.27	2.69 ± 2.60	2.80 ± 2.74	1.91 ± 2.46	1.52 ± 1.08	2.93 ± 2.06
*T_1/2e_* (h)	2.05 ± 1.11	2.28 ± 1.86	2.35 ± 1.44	2.89 ± 2.26	4.33 ± 1.31 *	3.21 ± 3.17
*T_1/2d_* (h)	2.14 ± 2.57	0.960 ± 0.734	1.36 ± 1.37	1.22 ± 1.24	2.33 ± 2.71	1.40 ± 1.74
*T_1/2a_* (h)	1.29 ± 1.25	0.529 ± 0.478	0.605 ± 0.546	1.57 ± 2.28	1.18 ± 1.22	0.357 ± 0.228
*C_max_* (ng/mL)	31.4 ± 16.3	77.0 ± 53.8	368 ± 253 ^aaa^	64.5 ± 14.7 ***	86.6 ± 31.0	755 ± 298 * ^bbb^
*T_max_* (h)	0.500 ± 0.267	0.365 ± 0.271	0.344 ± 0.186	0.500 ± 0.267	0.240 ± 0.129	0.448 ± 0.630
*AUC_(0-t)_* (ng*h/mL) × 10	7.49 ± 3.63	10.0 ± 4.48	50.2 ± 18.0 ^aaa^	14.1 ± 5.67 *	13.9 ± 4.06	106 ± 54.0 * ^bbb^
*AUC_(0-_**_∞)_*(ng*h/mL) × 10	8.31 ± 3.89	10.9 ± 4.79	58.7 ± 27.5 ^aaa^	15.8 ± 6.54 *	15.3 ± 4.31	111 ± 50.8 * ^bbb^
*MRT* (h)	3.36 ± 1.38	2.83 ± 1.41	3.17 ± 1.48	4.02 ± 1.96	3.28 ± 1.02	2.88 ± 1.93
*V_d_* (L/kg) × 10	80.3 ± 39.2	136 ± 105	51.4 ± 33.1	59.0 ± 35.1	172 ± 61.6	60.3 ± 89.6
*CL/F* (L/h/kg)	323 ± 206	431 ± 178	163 ± 70.4	158 ± 90.6	278 ± 68.5 *	91.2 ± 50.8 * ^bbb^

* *p* < 0.05; ** *p* < 0.01; *** *p* < 0.001 (Combined administration compared with single administration). ^aaa^
*p* < 0.001 (HAR-H compared with HAR-M). ^b^
*p* < 0.05; ^bbb^
*p* < 0.001 (HAR-H + MEM compared with HAR-M + MEM).

**Table 3 molecules-24-01430-t003:** Pharmacokinetics parameters of HOL in rats plasma after oral administration of medium (40.0 mg/kg) and high (80.0 mg/kg) doses of HAR and medium (40.0 mg/kg) and high (80.0 mg/kg) doses of HAR combined with MEM (5.0 mg/kg) (n = 8, mean ± SD).

Pharmacokinetics Parameters	HAR-M	HAR-H	HAR-M + MEM	HAR-H +MEM
*K_e_* (/h)	0.400 ± 0.133	0.226 ± 0.0896 ^aa^	0.312 ± 0.271	0.0899 ± 0.0912 ** ^b^
*K_d_*(/h)	0.433 ± 0.152	0.840 ± 0.948	0.966 ± 1.59	0.971 ± 1.10
*K_a_* (/h)	0.808 ± 1.28	1.17 ± 1.96	1.98 ± 1.72	0.864 ± 1.00
*T_1/2e_* (h)	1.94 ± 0.712	3.57 ± 1.50 ^a^	3.53 ± 2.19	17.2 ± 14.7 * ^b^
*T_1/2d_* (h)	1.89 ± 1.07	2.02 ± 1.72	2.05 ± 1.61	3.48 ± 4.35
*T_1/2a_* (h)	2.22 ± 1.67	2.02 ± 1.52	2.78 ± 4.59	2.15 ± 2.05
*C_max_* (ng/mL)	10.4 ± 2.13	32.4 ± 16.5 ^aa^	18.0 ± 5.95 *	40.4 ± 10.9 ^bbb^
*T_max_* (h)	0.198 ± 0.324	0.271 ± 0.159	0.229 ± 0.0589	2.39 ± 3.52
*AUC_(0-t)_* (ng*h/mL)	27.1 ± 10.8	57.8 ± 19.3 ^aaa^	35.5 ± 15.3	158 ± 59.9 ** ^bbb^
*AUC_(0-_**_∞)_*(ng*h/mL)	29.6 ± 11.3	69.7 ± 23.1 ^aaa^	43.1 ± 21.9	327 ± 140 *** ^bbb^
*MRT* (h)	4.44 ± 1.78	6.04 ± 2.12	5.52 ± 2.55	23.5 ± 19.3 * ^b^
*V_d_* (L/kg) × 10	463 ± 308	616 ± 225	499 ± 312	633 ± 462
*CL/F* (L/h/kg) × 10	186 ± 171	130 ± 56.7	107 ± 34.1	27.8 ± 9.14 *** ^bbb^

* *p* < 0.05; ** *p* < 0.01; *** *p* < 0.001 (Combined administration compared with single administration). ^a^
*p* < 0.05; ^aa^
*p* < 0.01; ^aaa^
*p* < 0.001 (HAR-H compared with HAR-M). ^b^
*p* < 0.05; ^bbb^
*p* < 0.001 (HAR-H + MEM compared with HAR-M + MEM).

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
