# Peer review of "Potential Pharmacokinetic Drug–Drug Interaction Between Harmine, a Cholinesterase Inhibitor, and Memantine, a Non-Competitive N-Methyl-d-Aspartate Receptor Antagonist"

_molecules, 2019, doi:10.3390/molecules24071430_

Round 1

Reviewer 1 Report

Based on the potential benefits of combining harmine (HAR), which displays effects of improving learning and memory, but may cause tremor related to the glutamate system; and memantine (MEM), which can also be used to treat Alzheimer’s disease and blocks glutamate neurotoxicty.  The authors posit that combination of these drugs may be clinically useful.  Using a rat model the authors test to see if a pharmacokinetic drug interaction may exist between the two drugs.  They find that when the two drugs are given together orally, the Cmax of MEM was decreased yet its AUC was increased; while the Cmax and AUC of HAR, and its metabolite, harmol (HOL) were increased, suggesting the combination would extend their resident times and increase treatment efficacy.  These are some encouraging results, but some issues are present in the manuscript, and limitations of the model must be discussed.

1.     Page 1, line 22-23 – As written implies assessment of HAR and HOL after administration of MEM alone.

2.     Page 1, lines 23 – Unless abstract word limits preclude, provide some information on doses and time of dosing.

3.     Page 1, line 33 (and throughout) – “shown” not “showed”.

4.     Page 1, line 37 – references 2 and 3 only indirectly show these usage patterns. Please provide some ‘direct’ references.

5.     Page 2, lines 44-46 – While references to the toxic mechanism of MTPT are appropriate here, also need references that show the toxicity of HAR is similar to MTPT.

6.     Page 2, line 68 – “In recent years” is not needed and can be deleted.

7.     Page 2, line 71 – Just “Pretreatment” not “The pretreatment”.

8.     Page 2, lines 75-76 – Just “pharmacokinetic” not “pharmacokinetics”.

9.     Introduction in general – Some references on pharmacokinetics of HAR and MEM should be provided and later discussed to show agreement (or not) with those reported here. (e.g., Ya et al. Pharmaceutics 10; (2018))

10.   Page 3, line 110 – Perhaps ‘matrix’ is better word than ‘substance’.

11.   Page 4, line 123 – ‘Concentrations levels’ is a reiterative phrase.  ‘Concentration’ suffices by itself, and in cases where ‘levels’ is used by itself, ‘concentrations’ is more accurate.

12.   Page 4, line 123 – the 1stuse of ‘400.00’ should be 100.00 

13.   Throughout numbers should be limited to 3 significant figures (in #12 100 not 100.00)

14.   Page 4, line 145 – “…at low, medium and high three QCs levels…” try “…at low, medium and high QCs concentrations…” (the ‘three’ is redundant) (also page 2nd 4 of 17, rows 195-196; and 2nd5 of 17 rows 204-205) 

15.   Page 7-9 – Table 1, 2 and 3 – Provide Vd and Cl as L/kg and L/h/kg to use smaller numbers.

16.   Page 8, Table 2 – perform stats between HAR-L, HAR-M and HAR-H, as well as HAR-L+MEM, HAR-M+MEM and HAR-H+MEM to show dose-dependency.

17.   Page ? (second use of 1 of 17) it follows Table 3 page 9 – we have a restart of page numbering. i.e., 2nd1 of 17.

18.   2nd1 of 17, row 7 – the “rose slowly” does not apply to all conditions

19.   2nd1 of 17, row 55 – I think ‘recirculation’ is more appropriate than ‘circulation’.

20.   2nd2 of 17, row 71-75 – The species of the transporters must be noted, and the relevance to this study in rats discussed as transporters show species specificity.

21.   2nd2 of 17, row 84 – While use CYP nomenclature for rats not humans.

22.   2nd2 of 17, row 102 – Do not use ‘etc’, be specific.

23.   2nd5 of 17, row 226 – ‘Mean ± SD’

24.   References – In. the text ref 42 and 43 were cited after ref 44.

Author Response

Dear Reviewer,

Thank you very much! We really appreciate you for your careful review for our manuscript titled of “Potential pharmacokinetic drug-drug interaction between harmine, a cholinesterase inhibitor, and memantine, a non-competitive N-methyl-D-aspartate receptor antagonist” and your consideration for publication it in Molecules. We have revised our manuscript according to your comments advised with highlighted color (RED).

Specific comments are responded in below:

Comments and Suggestions for Authors:

Based on the potential benefits of combining harmine (HAR), which displays effects of improving learning and memory, but may cause tremor related to the glutamate system; and memantine (MEM), which can also be used to treat Alzheimer’s disease and blocks glutamate neurotoxicity. The authors posit that combination of these drugs may be clinically useful. Using a rat model the authors test to see if a pharmacokinetic drug interaction may exist between the two drugs. They find that when the two drugs are given together orally, the Cmax of MEM was decreased yet its AUC was increased; while the Cmax and AUC of HAR, and its metabolite, harmol (HOL) were increased, suggesting the combination would extend their resident times and increase treatment efficacy. These are some encouraging results, but some issues are present in the manuscript, and limitations of the model must be discussed.

Response: Thank you very much! We really appreciate you for your careful review for our manuscript. Your professional advice makes this manuscript clearer and more sensible. For the limitations of the model, we have discussed the issues in manuscript as follow:

  The pharmacokinetic parameters may vary from species to species and the interspecies differences may origin from drug absorption, distribution, metabolism and elimination. Biotransformation is a major factor accounting for species differences in the disposition of drug [49]. Animal models, rats in particular, are commonly used to predict the metabolism and toxicity of new human drug candidates. However, it is important to realize that humans differ from animals in metabolic enzyme isoform composition, expression and catalytic activity and even small changes in the amino acid sequences of these enzymes can give rise to profound differences in substrate specificity and catalytic activity [50]. These magnify a complexity in the extrapolation of animal’s data to humans.

Comparative metabolism is relevant when identifying animal models for humans that have an appropriate similarity, but none of the investigated species matches all the typical CYP450 activities as described in humans [49]. With the aim of establishing the best animal model for human CYP450-related research, Li et al. undertake a comprehensive comparative investigation of the enzyme activities and kinetic parameters of HAR for individual CYP450 activities in liver microsomes or S9 fractions derived from rat, mouse, pig, bull, sheep, camel, rabbit, dog, monkey, guinea pig and human. The overall conclusion of this comparison is that none of the investigated species matches all the typical CYP450 activities as described in humans. And dogs and humans show considerable similarity in the metabolic profiles and catalytic processes of HAR [51]. It suggests that the pharmacokinetic drug-drug interaction between HAR and MEM would need to be carried out on other different animal models, such as dogs, monkeys and even clinical trials in the future study. This can be more comprehensive, can better guide clinical medication.” to the Discussion. (Now in 2nd3 of 18, line 123-143).

1. Page 1, line 22-23 – As written implies assessment of HAR and HOL after administration of MEM alone.

Response: Thanks for your comments! The sentence has been changed to “A UPLC-MS/MS method was established and validated for the simultaneous quantitative determination of MEM, HAR and harmol (HOL), a main metabolite of HAR, in rat plasma after oral administration of HAR and MEM in combination (5.0 mg/kg of MEM combined with 20.0, 40.0, 80.0 mg/kg of HAR). The contents of HAR and HOL were determined after oral administration of HAR (20.0, 40.0 and 80.0 mg/kg), and the content of MEM was determined after oral administration of MEM (5.0 mg/kg).” (Now in page 1, line 21-26).

2. Page 1, lines 23 – Unless abstract word limits preclude, provide some information on doses and time of dosing.

Response: Thank you very much! The doses and time of dosing have been added.(Now in page 1, line 26-27).

3. Page 1, line 33 (and throughout) – “shown” not “showed”.

Response: Thank you very much for your advice! We have checked the whole manuscript carefully and all grammar errors have been carefully corrected.

4. Page 1, line 37 – references 2 and 3 only indirectly show these usage patterns. Please provide some ‘direct’ references.

Response: Thanks a lot! We have modified references 2 and 3 to direct references in this manuscript.

5. Page 2, lines 44-46 – While references to the toxic mechanism of MTPT are appropriate here, also need references that show the toxicity of HAR is similar to MTPT.

Response: Thank you for your suggestion! New references have been added, which indicate that HAR is structurally similar to MPTP and show similar neurotoxicity.

6. Page 2, line 68 – “In recent years” is not needed and can be deleted.

Response: Thanks! It has been deleted.

7. Page 2, line 71 – Just “Pretreatment” not “The pretreatment”.

Response: We are very sorry for the unprofessional presentation, it has been revised.

8. Page 2, lines 75-76 – Just “pharmacokinetic” not “pharmacokinetics”.

Response: Thanks a lot, it has beenmodified.

9. Introduction in general – Some references on pharmacokinetics of HAR and MEM should be provided and later discussed to show agreement (or not) with those reported here. (e.g., Ya et al. Pharmaceutics 10; (2018)) 

Response: Thank you very much! We strongly agree with your suggestion that we have added the pharmacokinetic references of HAR and MEM in rats and discussed in Discussion (Now in page 2, line 76-81, 2nd2 of 18, line 77 and line 99-100).

10. Page 3, line 110 – Perhaps ‘matrix’ is better word than ‘substance’.

Response: Thanks, it has been amended.

11. Page 4, line 123 – ‘Concentrations levels’ is a reiterative phrase. ‘Concentration’ suffices by itself, and in cases where ‘levels’ is used by itself, ‘concentrations’ is more accurate.

Response: Thank you! We are very sorry for the unprofessional presentation, it has been revised.

12. Page 4, line 123 – the 1stuse of ‘400.00’ should be 100.00

Response: Thanks, it has been modified.

13. Throughout numbers should be limited to 3 significant figures (in #12 100 not 100.00)

Response: Thanks for your kindly suggestion. We have checked the whole manuscript carefully and throughout numbers have been limited to 3 significant figures.

14. Page 4, line 145 – “…at low, medium and high three QCs levels…” try “…at low, medium and high QCs concentrations…” (the ‘three’ is redundant) (also page 2nd 4 of 17, rows 195-196; and 2nd5 of 17 rows 204-205)

Response: Thank you! We have checked the whole manuscript carefully and all errors have been carefully corrected.

15. Page 7-9 – Table 1, 2 and 3 – Provide Vd and Cl as L/kg and L/h/kg to use smaller numbers.

Response: Thanks for your advice, it has been adjusted accordingly.

16. Page 8, Table 2 – perform stats between HAR-L, HAR-M and HAR-H, as well as HAR-L+MEM, HAR-M+MEM and HAR-H+MEM to show dose-dependency.

Response: Thanks for your comment! In order to better show dose-dependency, the stats have been conducted between HAR-L, HAR-M and HAR-H, as well as HAR-L+MEM, HAR-M+MEM and HAR-H+MEM (Now in 2nd1 of 18, line 18-20, line 27-28, line 31-35, line 39-40, line 46-47 and line 51-52).

17. Page? (Second use of 1 of 17) it follows Table 3 page 9 – we have a restart of page numbering. i.e., 2nd1 of 17.

Response: Thanks for your kindly reminder. We have realized that a restart of page numbering has been applied following Table 3 page 9. We also hope that the editor can notice this.

18. 2nd1 of 17, row 7 – the “rose slowly” does not apply to all conditions

Response: Thank you! For more rigorous expression, the sentence “the mean plasma concentration time curves of MEM rose slowly” has been changed to “the mean plasma concentration time curves of MEM rose slowly (in HAR-L+MEM and HAR-H+M groups)” (Now in 2nd1 of 18, line 7-10).

19. 2nd1 of 17, row 55 – I think ‘recirculation’ is more appropriate than ‘circulation’.

Response: Thanks, it has beenamended.

20. 2nd2 of 17, row 71-75 – The species of the transporters must be noted, and the relevance to this study in rats discussed as transporters show species specificity.

Response: Thank you very much! For the issue you pointed out, the species of the transporters have been given in the revised manuscript. These results may provide an important reference for the study of transporters in rats in later stage. The sentence “It is shown that HAR is a substrate for the absorption transporter OCTs/OCTNs and OATPs in human colon carcinoma and Madin-Darby canine kidney cells[32], and MEM is a substrate for the absorption transporter OCT2 and OATP2B1 in Madin-Darby canine kidney and human embryonic kidney cells [33,41] (Now in 2nd2 of 18, line 86-89). ” In addition, we have discussed these issues in discussion part of revised manuscript as follow:

In addition, drug transporters also play an important role in the absorption, distribution, and excretion of many drugs. There are significant differences between rodents, dog, monkey and human in the substrate specificity, tissue distribution, and relative abundance of transporters [55]. These differences complicate cross-species extrapolations, which is crucial when attempting to predict human pharmacokinetics of drug candidates and assess risk for drug-drug interactions. But, quantitative knowledge of species differences of transporters, especially at the protein and functional level is still limited. Further increased understanding of species differences in transporter expression and functional activity is needed in order to translate findings from preclinical species to humans, which will improve the capability to predict drug-drug interactions characteristics of drug candidates in humans (Now in 2nd3 of 18, line 149-158).

21. 2nd2 of 17, row 84 – While use CYP nomenclature for rats not humans.

Response: Thank you for your comment! The CYP metabolic enzymes are named differently in rats and humans. We have discussed these issues in discussion part of revised manuscript as follow:

By the way, it should be given a great attention that the nomenclature and function of CYP metabolic enzymes are different among different species. Such as, the amino acid sequences of CYP2A1 and CYP2A2 in rats and CYP2A6 in human are highly homologous [52], CYP2D22 of mice is homologous to CYP2D6 of humans [53], and CYP2D21 of pig is highly homologous to CYP2D6 of human [54](Now in 2nd3 of 18, line 144-148).” has been added to Discussion. In addition, the sentence “Studies have shown that HAR is mainly metabolized by CYP1A2, CYP2D6 and CYP3A4 in human and rat liver microsomes [42-44], and MEM is a strong inhibitor of CYP2D6, and a weak inhibitor of CYP1A2 and CYP3A4 in rat liver microsomes [45].

22. 2nd2 of 17, row 102 – Do not use ‘etc’, be specific.

Response: Thanks! It has been modified to “On the other hand, drug-drug interaction is a complex process and the combination of HAR and MEM may have a certain effect on changes in glutamate, dopamine, choline, inflammatory factors and reactive oxygen species.” (Now in 2nd3 of 17, line 116).

23. 2nd5 of 17, row 226 – ‘Mean ± SD’

Response: Thank you, it has been revised.

24. References – In. the text ref 42 and 43 were cited after ref 44.

Response: Thanks a lot, it has been revised.

Thanks again for your kind help. We hope this revised version of our manuscript will be acceptable for publication. For any other questions, please feel free to contact us and we will response at our first time.

Sincerely yours,

Chang-hong Wang Ph.D

Institute of Chinese Materia Medica, Shanghai University of Traditional Chinese Medicine.

1200 Cailun Road, Shanghai 201203, China

Tel: 86-21-51322511

Fax: 86-21-51322519

2019-04-05

Reviewer 2 Report

The authors have done a pioneering job and got a significant result for preliminary exploration of combinational use of HAR and MEM in the perspective of pharmacokinetics. In this manuscript, the authors has developed a UPLC-NM/MS method to simultaneously detect HAR,MEM and HAL as major metabolite in vivo with a good experimental design and also they have conducted a systematic comparison of the ADME parameters to draw a conclusion that co-administration of HAR and MEM can extend the residence time in rat which provided a basis for the feasibility of further study on how MEM can decrease the toxicity of HAR in combinational usage for AD treatment  clinically.

Author Response

Dear Reviewer,

Thank you very much! We really appreciate you for your careful review for our manuscript titled of “Potential pharmacokinetic drug-drug interaction between harmine, a cholinesterase inhibitor, and memantine, a non-competitive N-methyl-D-aspartate receptor antagonist” and your consideration for publication it in Molecules.

Specific comments are responded in below:

Comments and Suggestions for Authors:

The authors have done a pioneering job and got a significant result for preliminary exploration of combinational use of HAR and MEM in the perspective of pharmacokinetics. In this manuscript, the authors has developed a UPLC-NM/MS method to simultaneously detect HAR, MEM and HAL as major metabolite in vivo with a good experimental design and also they have conducted a systematic comparison of the ADME parameters to draw a conclusion that co-administration of HAR and MEM can extend the residence time in rat which provided a basis for the feasibility of further study on how MEM can decrease the toxicity of HAR in combinational usage for AD treatment clinically.

Response: Thank you very much! We really appreciate you for your careful review for our manuscript. Thank you for your review of our work, we will continue to work harder and strive to do more perfect. Your support has provided a lot of motivation for our research work, thank you very much!

Thanks again for your kind help. We hope this revised version of our manuscript will be acceptable for publication. For any other questions, please feel free to contact us and we will response at our first time.

Sincerely yours,

Chang-hong Wang Ph.D

Institute of Chinese Materia Medica, Shanghai University of Traditional Chinese Medicine.

1200 Cailun Road, Shanghai 201203, China

Tel: 86-21-51322511

Fax: 86-21-51322519

2019-04-05

Reviewer 3 Report

The main issue of this study was the elucidation of pharmacokinetic drug-drug interaction between memantine and harmine. This manuscript was well designed and the authors obtained meaningful results. However, there are several issues that the authors need to revise before the manuscript can be accepted for publication.

Major comments

1. In the disccuion section, authors speculated that the possible mechanisms for pharmacokinetic interaction between memantine and harmine might be the inhibition in urinary excretion process, the competition to absorption transporters and metabolic inhibition. Therefore, these possibilities mentioned in discussion section should be evaluated through further experiments such as ussing chamber study using rat intestine, metabolic inhibition study and transport study for OCT-expression cell etc.  

2. In Figure 2, the harmine concentration in HAR-H+MEM group seemed to be beyond the calibration ranges. This issue should be explained in terms of method validation

3. Table 1-3, There were errors in calculation of CL. Because only PO study was conducted in this manuscript, absolute bioavailability (F) value was unknown in this case. Therefore, “CL” in table 1-3 should be corrected to “CL/F”   

Monor comments
1. Please Insert the product ion spectra of MEM, HAR, and HOL in the manuscript.

Author Response

Dear Reviewer,

Thank you very much for your comments on our manuscript titled of “Potential pharmacokinetic drug-drug interaction between harmine, a cholinesterase inhibitor, and memantine, a non-competitive N-methyl-D-aspartate receptor antagonist” and your consideration for publication it in Molecules. We have revised our manuscript according to your comments advised with highlighted color (RED).

Specific comments are responded in below:

Comments and Suggestions for Authors:

The main issue of this study was the elucidation of pharmacokinetic drug-drug interaction between memantine and harmine. This manuscript was well designed and the authors obtained meaningful results. However, there are several issues that the authors need to revise before the manuscript can be accepted for publication.

Response: Thank you very much for your comments! We really appreciate you for your careful review for our manuscript. Your professional advice makes this manuscript clearer and more sensible.

Major comments

1. In the discussion section, authors speculated that the possible mechanisms for pharmacokinetic interaction between memantine and harmine might be the inhibition in urinary excretion process, the competition to absorption transporters and metabolic inhibition. Therefore, these possibilities mentioned in discussion section should be evaluated through further experiments such as using chamber study using rat intestine, metabolic inhibition study and transport study for OCT-expression cell etc.

Response: Thank you very much for your suggestion! In order to deeply verify the possible mechanisms for pharmacokinetic interaction between memantine and harmine, further experiments will be necessary such as using chamber study using rat intestine, metabolic inhibition study and transport study for OCT-expression cell etc.

2. In Figure 2, the harmine concentration in HAR-H+MEM group seemed to be beyond the calibration ranges. This issue should be explained in terms of method validation

Response: Thanks very much! The plasma was diluted 6.25 times during this experiment. So, the concentration of harmine in HAR-H+MEM group was within the range of the calibration. Moreover, the dilution integrity of harmine has been well-validated in our previous study (Li, S.P.; Zhang, Y.P.; Deng, G.; Wang, Y.W.; Qi, S.L.; Cheng, X.M.; Ma, .YM.; Xie, Y.; Wang, C.H. Exposure characteristics of the analogous β-carboline alkaloids harmaline and harmine based on the efflux transporter of multidrug resistance protein 2. Front. Pharmacol. 2017, 8, 541.).

3. Table 1-3, there were errors in calculation of CL. Because only PO study was conducted in this manuscript, absolute bioavailability (F) value was unknown in this case. Therefore, “CL” in table 1-3 should be corrected to “CL/F”.

Response: Thanks a lot! In this manuscript, all the pharmacokinetic parameters were processed using the non-compartmental pharmacokinetics data analysis software program PK solutions 2.0TM (Summit Research Services, USA), and the software can directly give the value of CL. CL = Ke ×Vd= Ke ×X/C (Ke is elimination rate constant; Vd is apparent volume of distribution; X is the dose to be administered; C is the concentration of the drug in plasma).

Minor comments

1. Please insert the product ion spectra of MEM, HAR, and HOL in the manuscript.

Response: Thank you very much! The product ion mass spectra of HAR, MEM, HOL and IS has been provided in revised manuscript as Figure 2.

Thanks again for your kind help. We hope this revised version of our manuscript will be acceptable for publication. For any other questions, please feel free to contact us and we will response at our first time.

Sincerely yours,

Chang-hong Wang Ph.D

Institute of Chinese Materia Medica, Shanghai University of Traditional Chinese Medicine.

1200 Cailun Road, Shanghai 201203, China

Tel: 86-21-51322511

Fax: 86-21-51322519

2019-04-05

Round 2

Reviewer 3 Report

1. In the equation for CL presented by the authors, that is, CL = Ke ×Vd= Ke ×X/C, the X means the absorbed amount in the body not simply the dose to be administered, therefore, Vd=X/C is just right in the case of iv bolus administration because of absolute bioavailability is 100%.

Therefore, the CL in the manuscript should be changed to the CL/F

Author Response

Dear Reviewer,

Thank you very much! We really appreciate you for your careful review for our manuscript titled of “Potential pharmacokinetic drug-drug interaction between harmine, a cholinesterase inhibitor, and memantine, a non-competitive N-methyl-D-aspartate receptor antagonist” and your consideration for publication it in Molecules. We have revised our manuscript according to your comments advised with with Track Changes function in Microsoft Word.

Specific comments are responded in below:

Comments and Suggestions for Authors:

1. In the equation for CL presented by the authors, that is, CL = Ke ×Vd= Ke ×X/C, the X means the absorbed amount in the body not simply the dose to be administered, therefore, Vd=X/C is just right in the case of iv bolus administration because of absolute bioavailability is 100%.

Therefore, the CL in the manuscript should be changed to the CL/F.

Response: Thanks for your kindly suggestion! You are right! The CL in the manuscript has been changed to the CL/F.

Thanks again for your kind help. We hope this revised version of our manuscript will be acceptable for publication. For any other questions, please feel free to contact us and we will response at our first time.

Sincerely yours,

Chang-hong Wang Ph.D

Institute of Chinese Materia Medica, Shanghai University of Traditional Chinese Medicine.

1200 Cailun Road, Shanghai 201203, China

Tel: 86-21-51322511

Fax: 86-21-51322519

2019-04-09